# In-situ self-assembly of hole transport monolayer during crystallization for efficient single-crystal perovskite solar cells

Vishal Yeddu [1,6], Khulud Almasabi[2,6], Yafeng Xu [2], Augusto Amaro[1], Shuang Qiu[1], Sergey Dayneko [1], Dongyang Zhang[1], Parinaz Moazzezi[3], Christopher Tremblay[1,4], Muhammad Naufal Lintangpradipto[2], Heather L. Buckley[1,4,5], Omar F. Mohammed [2], Osman M. Bakr [2,7] & Makhsud I. Saidaminov [1,3,4,7]

Single-crystal perovskite solar cells (SC-PSCs) are emerging as a promising technology owing to their intrinsically low defect densities, long carrier diffusion lengths, and enhanced stability compared to their polycrystalline counterpart. However, their performance has been limited by interface-related losses, particularly at the perovskite/charge transport layer, which hinders effective hole extraction and promotes non-radiative recombination. In this work, we introduce a self-assembled monolayer (SAM) deposition strategy that exploits an asymmetric substrate stack configuration during space-confined inverse temperature crystallization (SC-ITC). This configuration triggers an in-situ migration of SAM molecules from the SAM-coated substrate to the uncoated substrate, resulting in a denser and more homogeneous SAM coating than the conventional spin-coating method can achieve. The improved SAM coverage significantly enhances hole extraction. Consequently, our SC-PSCs achieved power conversion efficiency as high as 24.32%.

Polycrystalline perovskite solar cells (PSCs) have demonstrated power conversion efficiencies (PCEs) approaching 27%, thanks to significant progress in defect passivation and interface engineering[1–12]. Nevertheless, the inherent grain boundaries in polycrystalline films introduce electronic defects and structural instabilities[13–18], hindering charge transport and increasing vulnerability to environmental degradation from factors like moisture and oxygen[19–22]. In contrast, single-crystal perovskites eliminate grain boundaries altogether, resulting in notably lower trap densities, extended carrier diffusion lengths, and enhanced chemical and thermal stability[23–31]. Consequently, single-crystal perovskite solar cells (SC-PSCs) are gaining

traction as a promising evolution in photovoltaic technology, poised to address critical challenges that persist in traditional polycrystalline PSCs.

The current highest reported PCE for SC-PSCs stands at 25.8%, which is still lower than the peak efficiency achieved by polycrystalline PSCs[32]. The primary limitation hindering SC-PSC performance is the high surface defect density, which leads to increased recombination at the interfaces between the perovskite and charge transport layers[15,33]. Additionally, the poor adhesion of perovskite crystals to the substrate further reduces PCE[34]. Poor adhesion typically causes device failure, as the perovskite crystal tends to delaminate under operational

[1]Department of Chemistry, University of Victoria, 3800 Finnerty Road, Victoria, BC V8P 5C2, Canada. [2]Center for Renewable Energy and Storage Technologies (CREST), Division of Physical Science and Engineering, King Abdullah University of Science and Technology (KAUST), Thuwal 23955-6900, Kingdom of Saudi Arabia. [3]Department of Electrical & Computer Engineering, University of Victoria, 3800 Finnerty Road, Victoria, BC V8P 5C2, Canada. [4]Center for Advanced Materials and Related Technologies (CAMTEC), University of Victoria, Victoria, BC V8P 5C2, Canada. [5]Department of Civil Engineering, University of Victoria, 3800 Finnerty Road, Victoria, BC V8P 5C2, Canada. [6]These authors contributed equally: Vishal Yeddu, Khulud Almasabi. [7]These authors jointly supervised this work: Osman M. Bakr, Makhsud I. Saidaminov. ✉e-mail: osman.bakr@kaust.edu.sa; msaidaminov@uvic.ca

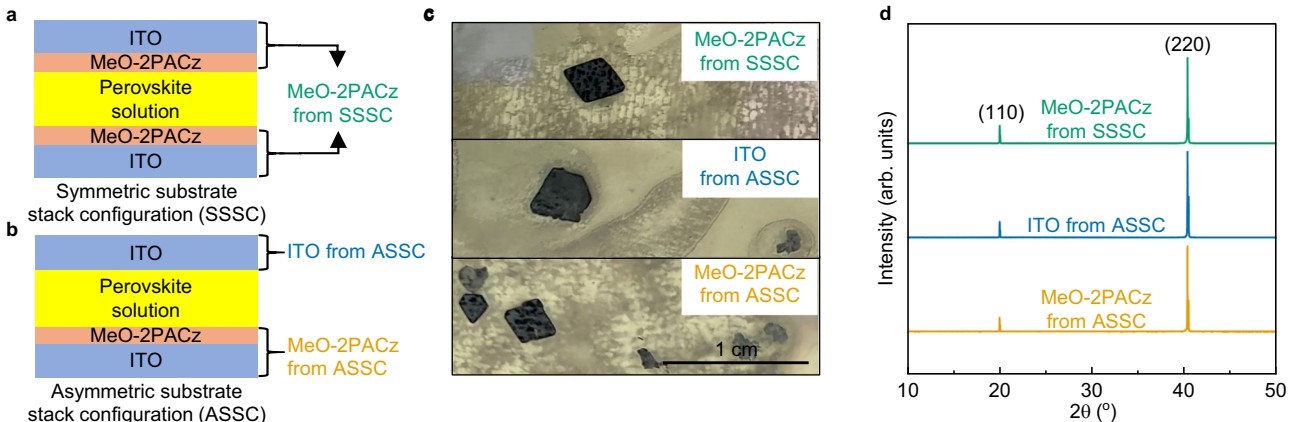

**Fig. 1 | Symmetric/asymmetric substrate stack configurations. a** Schematic of symmetric and **b** asymmetric substrate stack configurations used during SC-ITC. **c** Images of obtained perovskite crystals. **d** XRD pattern of the perovskite crystals. Source data for the plots are provided as a Source Data file.

conditions[34]. Self-assembled monolayers (SAMs) have been utilized as hole transport layer (HTL) to address these challenges. SAMs not only enhance the adhesion between perovskite crystals and the substrate but also passivate surface defects on the perovskite layer[34]. As a result, the incorporation of SAMs has significantly improved the PCE of SC-PSCs and greatly enhanced their operational stability[34].

Carbazole derivatives with phosphonic acid anchoring groups (PACz series) are the most widely used SAMs in PSCs[35–37]. These SAMs offer favorable optoelectronic properties for serving as HTL in PSCs, including high optical transmittance[34], interface defect passivation[38], improved hole extraction via electrode work function modulation[39], low parasitic absorption[40], and fast charge transfer rate[35,37,39]. However, there remain certain limitations associated with SAM processing, particularly because these SAM molecules are amphiphilic in nature and tend to form micelles in solvents commonly used to dissolve them, hindering their effective interaction with transparent conductive oxide electrodes such as indium tin oxide (ITO)[41,42]. As a result, the SAM molecules tend to form clusters on the ITO surface instead of a monolayer[41,42]. Moreover, anchoring of SAM molecules to the ITO can be compromised by polar solvents like N,N′-dimethylformamide (DMF), leading to SAM desorption[43]. Together, these issues lead to incomplete and non-uniform coverage of ITO by SAM molecules, resulting in reduced performance due to ineffective hole extraction, increased leakage current, and poor perovskite attachment caused by reduced wettability[41]. Thus, improving coverage and achieving a uniform distribution of SAM molecules is necessary for minimizing interface defects causing non-radiative recombination, enhancing the PCE of PSCs.

In this work, we demonstrate that using an ITO/SAM asymmetric substrate stack during space-confined inverse temperature crystallization (SC-ITC) can yield in-situ formation of a dense and uniform SAM layer on bare ITO. Remarkably, SC-PSCs utilizing these in-situ functionalized ITO substrates achieve a champion PCE of 24.32% which is amongst the highest reported efficiencies for cesium-free perovskite-based SC-PSCs. We show that SAM molecules can desorb from the SAM-coated substrate and re-adsorb onto the initially uncoated ITO, improving molecular packing and interfacial properties. By employing various surface characterization techniques, we confirm this in-situ deposition process and reveal a higher surface density of SAM on the re-adsorbed layer than on conventional spin-coated films. The enhanced packing leads to more efficient hole extraction, reduced charge recombination, and improved device stability. These findings reveal a powerful strategy for overcoming the coverage limitations inherent to SAM-based HTLs and underscore the crucial role of substrate stack design in advancing the efficiency and stability of single-crystal perovskite photovoltaics.

## Results

### Perovskite crystallization using symmetric and asymmetric substrate stack configurations

In this study, SC-PSCs were fabricated using formamidinium$_{0.6}$ methylammonium$_{0.4}$ lead triiodide (FA$_{0.6}$MA$_{0.4}$PbI$_3$) perovskite crystal as an active layer and [2-(3,6-Dimethoxy-9H-carbazol-9-yl)ethyl]phosphonic acid (MeO-2PACz) SAM as the HTL. To fabricate solar cells using a perovskite single crystal as the active layer, studies have indicated that the optimal crystal thickness for achieving high PCE is around 25 μm[27,44,45]. To grow such thin single crystals, we employed a SC-ITC method[46,47]. This technique restricts crystal growth in the axial direction by performing the crystallization between HTL-coated ITO substrates. Figure 1a provides a schematic illustration of the substrate stack used in the SC-ITC. To date, all SC-PSCs fabricated using the SC-ITC method have utilized a symmetric substrate stack configuration (SSSC), in which both the bottom and top ITO substrates are coated with an HTL, MeO-2PACz in our case (Fig. 1a). However, here, we explored an asymmetric substrate stack configuration (ASSC), where one ITO substrate was coated with MeO-2PACz while the other was not (Fig. 1b). The primary objective was to investigate how the substrate stack configuration used during the crystallization process influences the performance of SC-PSCs. Details of the SC-ITC used for perovskite crystal growth are provided in the methods section.

The top panel of Fig. 1c shows the largest FA$_{0.6}$MA$_{0.4}$PbI$_3$ single crystal obtained in the current study using the MeO-2PACz SSSC, measuring 8.76 mm². For the ITO/MeO-2PACz ASSC, the largest crystal on the MeO-2PACz-coated ITO substrate measured 7.24 mm². Interestingly, the largest crystal grown on an uncoated ITO substrate from the ASSC was 12 mm², surpassing those obtained with MeO-2PACz-coated ITO substrates in both configurations. Supplementary Fig. 1 presents the statistical distribution of perovskite crystal sizes across different substrate types. Crystals grown on the ITO exhibited a larger average size compared to those on ITO/MeO-2PACz in both SSSC and ASSC configurations. These results suggest that, compared to MeO-2PACz, ITO provides a more favorable surface for FA$_{0.6}$MA$_{0.4}$PbI$_3$ single crystal growth.

The perovskite crystals shown in Fig. 1c appear to have spots on surfaces which we attribute to be defects resulting from residual perovskite growth solution left on the crystal after crystallization was terminated (Supplementary Fig. 2). As this residual solution cooled rapidly, it lost supersaturation, leading to localized dissolution of the perovskite. The use of surfactants such as cetyltrimethylammonium chloride has been shown to prevent such surface damage[44]. However, we chose not to use this surfactant in this work to solely focus on the effects of substrate stack configurations.

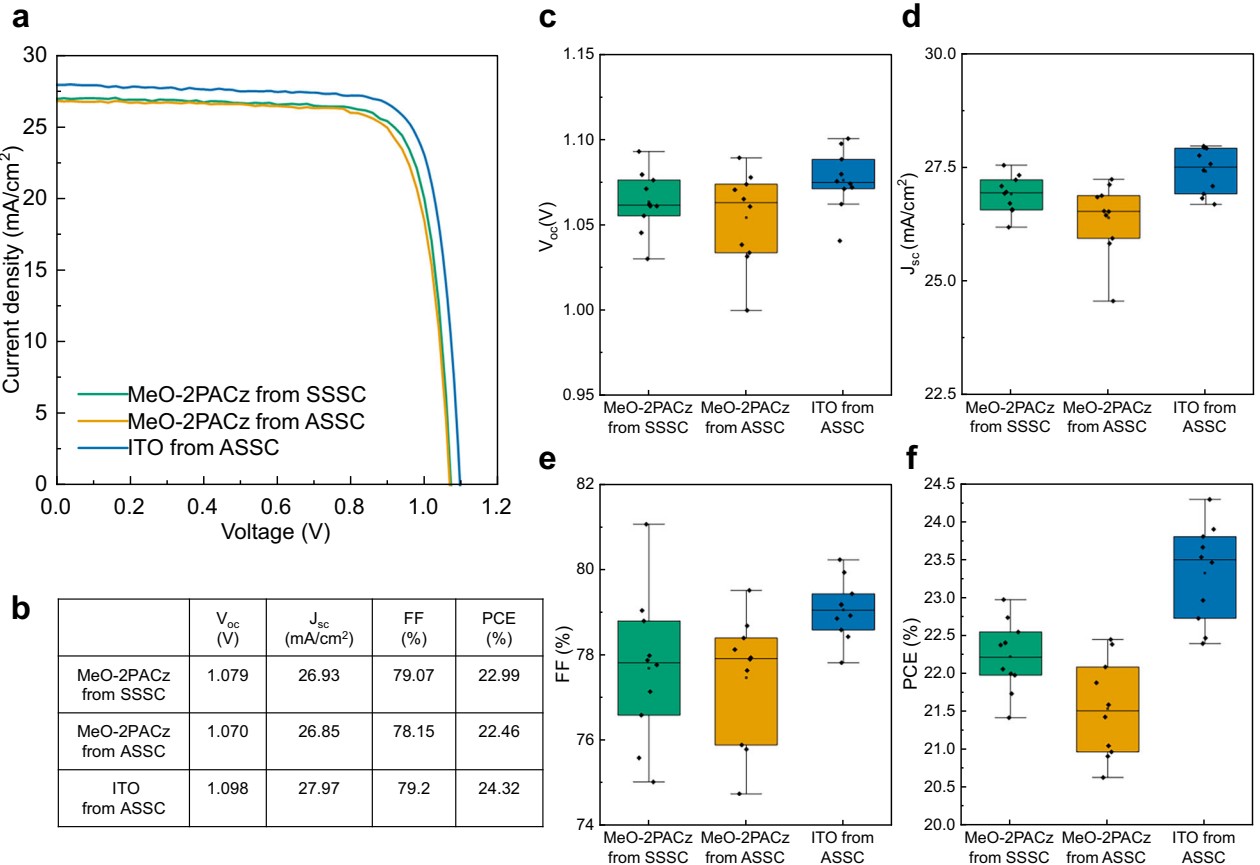

| | $V_{oc}$ (V) | $J_{sc}$ (mA/cm$^2$) | FF (%) | PCE (%) |
|---|---|---|---|---|
| MeO-2PACz from SSSC | 1.079 | 26.93 | 79.07 | 22.99 |
| MeO-2PACz from ASSC | 1.070 | 26.85 | 78.15 | 22.46 |
| ITO from ASSC | 1.098 | 27.97 | 79.2 | 24.32 |

**Fig. 2 | SC-PSCs performance. a** Reverse scan J-V characteristics of champion SC-PSCs obtained for each substrate type. **b** Table summarizing the key performance parameters of the champions cells. Statistical (**c**) $V_{oc}$, **d** $J_{sc}$, **e** FF, and **f** PCE data for 10 SC-PSCs on each type of substrate. In the box plots, the top and bottom edges of each box indicate the first and third quartiles (the 25$^{th}$ and 75$^{th}$ percentiles). This illustrates the spread of the middle half of the data. The line inside the box is the median line, highlighting the data's central tendency. The whiskers extending above and below each box show the range of values that fall within the normal spread. Source data for the plots are provided as a Source Data file.

The X-ray diffraction pattern of the crystals (Fig. 1d) exhibited only two peaks corresponding to (110) and (220) crystallographic planes, confirming the single-crystalline nature of the crystals grown on MeO-2PACz substrates in SSSC and ASSC, as well as the uncoated ITO substrate.

## Performance of SSSC and ASSC SC-PSCs

Solar cells were fabricated using these single crystals by deposition of C$_{60}$/bathocuproine (BCP)/Copper (Cu) on top of the single crystal surface. The fabrication process is discussed in more detail in the methods section. Figure 2a presents the reverse scan current density-voltage (J-V) characteristics for our champion devices. The open-circuit voltage ($V_{oc}$), short-circuit current density ($J_{sc}$), fill factor (FF), and PCE of control SC-PSC grown using SSSC were 1.079 V, 26.93 mA/cm$^2$, 79.07%, and 22.99%, respectively (Fig. 2b). To our surprise, the best performance was obtained for crystals grown on uncoated ITO from ASSC with a $V_{oc}$ of 1.098 V, $J_{sc}$ of 27.97 mA/cm$^2$, FF of 79.2%, resulting in a PCE of 24.32%. The forward and reverse scan J-V characteristics, as well as the EQE curve for this champion device, are shown in Supplementary Fig. 3a and Supplementary Fig. 3b respectively. Another unexpected result was that the PCE of the SC-PSC fabricated using crystals on the MeO-2PACz-coated substrate from the ASSC was lower than that of SC-PSCs fabricated using crystals on MeO-2PACz substrate from SSSC as well as crystals on ITO from ASSC. Figure 2c-f shows the statistical distribution of the solar cell parameters of all types of SC-PSCs we fabricated: it is apparent that SC-PSCs on ITO from ASSC have the highest PCE.

It appears that HTL-free SC-PSCs have the best performance, which is counter-intuitive. To verify whether HTL-free SC-PSCs are indeed more efficient, we fabricated devices directly on ITO using an ITO/ITO SSSC, as shown in Supplementary Fig. 4a−f. However, these devices exhibited low efficiency, with an average PCE of approximately 7.5% (Supplementary Fig. 4f) and a champion cell PCE of only 11.35% (Supplementary Fig. 5a, b). This indicates that HTL-free SC-PSCs fabricated using crystals grown on ITO cannot achieve high PCE. This outcome is expected as MeO-2PACz modifies the work function of ITO, enhancing hole collection compared to bare ITO[38]. The $V_{oc}$ for the symmetric ITO devices is significantly lower than that of asymmetric ITO devices. This is because of the large difference between the work function of ITO and perovskite valence band energy levels[48,49]. Furthermore, the hump observed in the J-V curve (Supplementary Fig. 5a) indicates charge accumulation in symmetric ITO devices[50]. Thus, our results confirm that incorporating an HTL is essential for achieving high-performance SC-PSCs using FA$_{0.6}$MA$_{0.4}$PbI$_3$ single crystals grown via SSSC. Additionally, this experiment highlights that the substrate stack used during perovskite crystal growth significantly impacts SC-PSCs performance.

## Investigation of ITO from ASSC

To understand the improved performance of ASSC ITO SC-PSCs, we took a closer look at their key solar cell parameters. The $V_{oc}$ and $J_{sc}$ of ASSC ITO SC-PSCs are significantly higher than that of SSSC ITO SC-PSCs but comparable to both SSSC and ASSC MeO-2PACz SC-PSCs. It suggests that the ITO substrate might be coated with MeO-2PACz

during the crystallization process. It has been reported that SAM molecules, particularly those weakly bonded to the substrate, tend to desorb when exposed to polar solvent molecules[43]. This effect is exacerbated at elevated temperatures and prolonged exposure times, conditions that are inherent to the SC-ITC process[51]. We, thus, hypothesized that in the ASSC, the desorbed SAM molecules reabsorb onto the initially uncoated ITO substrate during the crystallization process, thereby modifying the interfacial properties of the ITO. Such an unintended yet beneficial redistribution of SAM molecules could explain the enhanced photovoltaic performance observed in ASSC ITO SC-PSCs.

Figure 3a illustrates the molecular structure of MeO-2PACz. To verify its in-situ deposition on the asymmetric ITO surface, we removed the perovskite single crystal and performed X-ray photoelectron spectroscopy (XPS) at the location where the crystal was previously attached. For comparison, XPS measurements were also taken from bare ITO substrate and an ITO substrate spin-coated with MeO-2PACz. The XPS spectra in Fig. 3b reveal the presence of phosphorus on the ASSC ITO substrate, indicating the presence of MeO-2PACz.

We then conducted Fourier Transformed Infrared (FTIR) Spectroscopy to identify molecular vibrational modes characteristic of MeO-2PACz. The FTIR spectrum for the ASSC ITO substrate closely resembled that of the spin-coated MeO-2PACz reference sample (Fig. 3c) and aligns well with previously reported spectra in the literature[38]. Specifically, the bands observed near 1490-1494 cm$^{-1}$ and 1466-1483 cm$^{-1}$ correspond to the characteristic carbazole ring stretching vibrations of MeO-2PACz[38]. Additionally, the band at 1581 cm$^{-1}$ was attributed to the asymmetric stretching vibration of rings with adjacent methoxy groups[38]. None of these bands appear in the bare ITO FTIR spectrum. Combined, the XPS and FTIR results confirm the successful in-situ deposition of MeO-2PACz while using ASSC during SC-ITC.

However, these results do not explain the performance improvement we observed for the ASSC ITO devices, which, we now know, have in-situ deposited MeO-2PACz coating. To explain this improvement, we performed cyclic voltammetry (CV) measurements to determine the surface density of the MeO-2PACz molecules on the ITO surface. The details of the CV measurement, along with the calculations involved, are described in Supplementary Fig. 6a-6f. The MeO-2PACz surface density on MeO-2PACz coated substrate from SSSC, MeO-2PACz coated substrate from ASSC, and ITO substrate from ASSC was estimated to be $2.69\times10^{13}$, $2.31\times10^{13}$ and $3.52\times10^{13}$ molecules cm$^{-2}$, respectively. Interestingly, the highest surface density was obtained for in-situ MeO-2PACz coating. High surface density indicates improved MeO-2PACz packing, resulting in denser and uniform coating, enhancing the surface coverage. Increased SAM coverage helps improve the hole collection at the perovskite interface, which explains the boost in $J_{sc}$ and FF. Moreover, better coverage leads to lower leakage current, which boosts the $V_{oc}$.

To gain insight into the in-situ deposition process, we performed density functional theory (DFT) calculations to examine the interaction between MeO-2PACz and ITO. The XRD pattern of our ITO showed highest diffraction intensity of (222) crystallographic plane, which is a higher-order diffraction of (111) plane (Supplementary Fig. 7). Therefore, we modeled the InO-terminated ITO (111) surface. The results indicate that MeO-2PACz attaches to ITO (111) via two types of bonds: a P=O−In bond and a P−O−In bond (Fig. 3d). The latter P−O−In bond forms when a phosphonic acid O−H group in MeO-2PACz dissociates, yielding a deprotonated oxygen (O$^-$) that binds to an indium atom and a proton (H$^+$) that bonds to an oxygen on the ITO surface. We further calculated binding energies at 373.15 K (100 °C), 383.15 K (110 °C), 393.15 K (120 °C), and 403.15 K (130 °C) (spanning the typical substrate temperatures during perovskite crystallization). Figure 3e shows that the binding energy of MeO-2PACz with ITO decreases as the

temperature increases. During the SC-ITC process, the bottom substrate (in contact with the hot plate) is at a higher temperature than the top substrate (exposed to the cooler atmosphere, Supplementary Fig. 8). As a result, MeO-2PACz is expected to bind more strongly to the cooler top ITO surface than to the hotter bottom ITO.

Figure 3f illustrates a possible mechanism for the transfer of MeO-2PACz during the crystallization process. The concentration of MeO-2PACz in the ethanol used for spin-coating exceeds the critical micelle formation concentration, causing the molecules to form micelles in the solution; these aggregated structures remain intact after spin-coating, leading to non-uniform coverage and reduced ITO surface coverage[41]. During SC-ITC using an ITO/MeO-2PACz ASSC, the loosely bound MeO-2PACz molecules dissolve into the perovskite solution[43]. However, since the binding energy of MeO-2PACz with cooler top ITO substrate is higher than that of the hotter bottom ITO substrate, the dissolved MeO-2PACz molecules preferentially adsorb onto the top uncoated ITO substrate resulting in in-situ deposition. As this process is slower than spin-coating, in-situ deposition process during crystallization likely allows the MeO-2PACz molecules to organize into a more orderly monolayer, resulting in improved surface coverage.

Comparing the MeO-2PACz coverage for different substrates in SSSC and ASSC provides further insights into the in-situ deposition process. In SSSC, a greater total amount of MeO-2PACz is available to dissolve into the perovskite solution, since both top and bottom MeO-2PACz coated substrates contribute loosely bound SAM molecules. This higher dissolved concentration leads to more re-adsorption and slightly enhanced overall coverage on the spin-coated substrate in the SSSC, compared to the spin-coated substrate in ASSC. Interestingly, MeO-2PACz coverage on the top ITO substrate from ASSC exceeds that of the spin-coated substrates used in SSSC. This is probably because the spin-coating results in defects which cannot be healed by re-adsorption of MeO-2PACz dissolved in perovskite solution thereby limiting its surface coverage. In contrast, the initially uncoated ITO surface in ASSC enables fresh, more ordered adsorption of MeO-2PACz during crystallization, resulting in a more uniform and defect-free monolayer.

## Understanding the origin of enhanced performance in in-situ deposited MeO-2PACz SC-PSCs compared to spin-coated SC-PSCs

We investigated the differences between spin-coated MeO-2PACz and in-situ deposited MeO-2PACz by contact angle measurements using a supersaturated $FA_{0.6}MA_{0.4}PbI_3$ perovskite solution in gamma-butyrolactone (Supplementary Fig. 9a, c). The bare ITO substrate had a contact angle of 62.5° suggesting poor wettability of perovskite solution. On the other hand, the contact angle for spin-coated and in-situ deposited MeO-2PACz substrates were sharp, 13.4° and 11.3°, respectively. The lower contact for in-situ MeO-2PACz indicates better perovskite solution wettability compared to spin-coated MeO-2PACz. The improved wettability is due to enhanced surface coverage obtained using the in-situ deposition technique. The increased wettability also explains why crystals on ITO from ASSC were, on average, larger compared to crystals on MeO-2PACz from SSSC since better wettability can promote crystal growth[34].

To gain deeper insight into the enhanced performance of in-situ deposited MeO-2PACz SC-PSCs over spin-coated MeO-2PACz SC-PSCs, we investigated the hole transfer characteristics at the MeO-2PACz/perovskite interface. Figure 4a presents the steady-state photoluminescence (PL) spectra collected upon excitation from the ITO side of the devices as shown in Fig. 4a inset. The perovskite crystal grown on in-situ deposited MeO-2PACz exhibited a lower PL intensity compared to those grown on spin-coated MeO-2PACz, indicating stronger PL quenching by the in-situ deposited layer, suggesting more efficient hole extraction than the spin-coated layer[41,52–54].

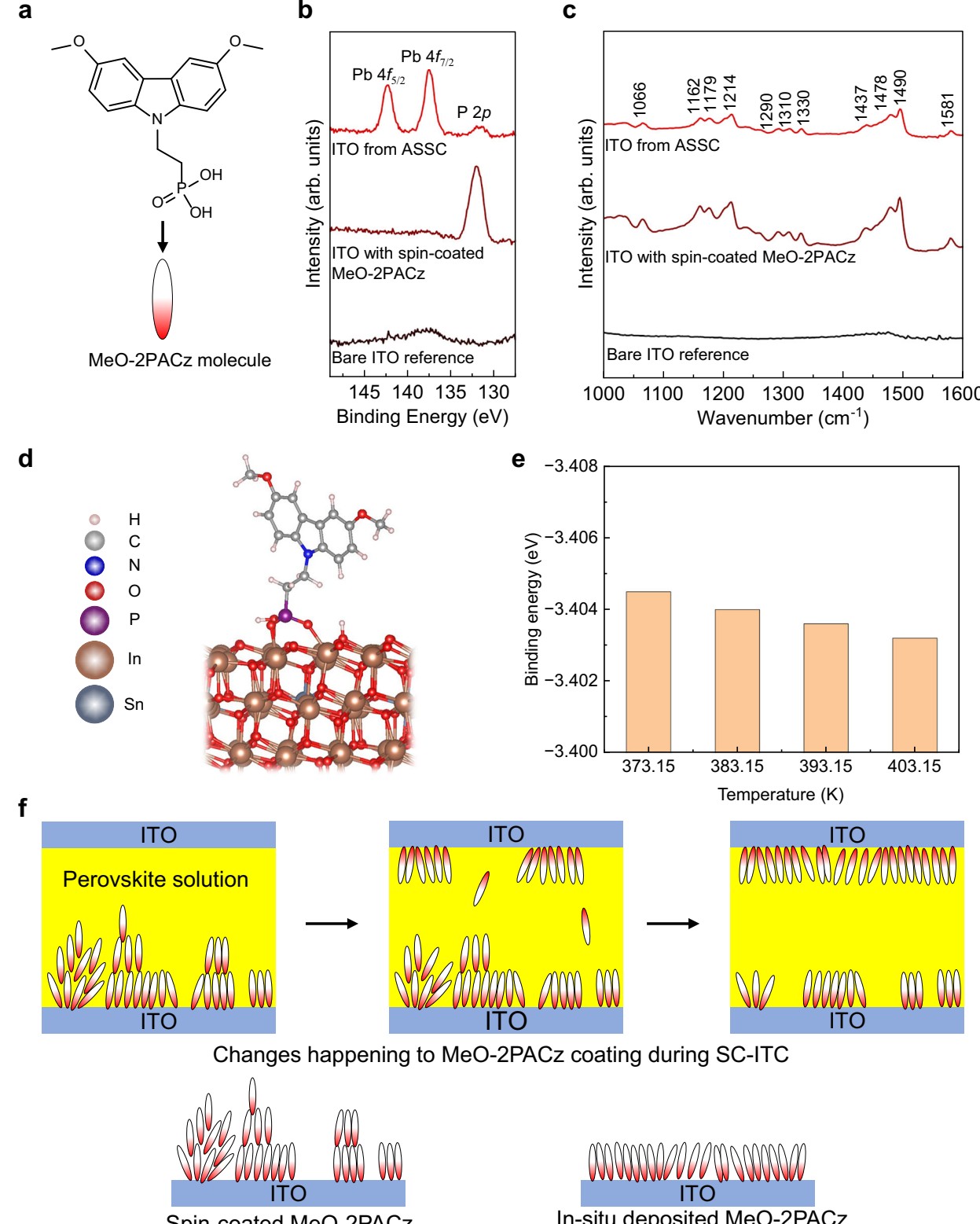

**Fig. 3 | Characterization of stacks. a** Molecular structure of MeO-2PACz. **b** XPS spectra of different substrates. **c** FTIR spectra of different substrates. **d** Side-view DFT-simulated schematic illustrating the anchoring of MeO-2PACz on InO-terminated ITO (111) surface. **e** ITO/MeO-2PACz binding energies at different temperatures. **f** Schematic illustrations of the in-situ MeO-2PACz deposition process. Source data for the plots are provided as a Source Data file.

Time-resolved PL (TRPL) measurements further supported these findings (Fig. 4b). The average PL lifetimes ($\tau_{average}$) of perovskite crystals deposited on bare ITO, spin-coated MeO-2PACz, and in-situ deposited MeO-2PACz were determined to be 5.06 ns, 1.84 ns, and 0.70 ns, respectively. The shortened $\tau_{average}$ in devices employing in-situ deposited MeO-2PACz highlights the enhanced hole extraction efficiency and faster charge transfer dynamics.

We performed atomic force microscopy (AFM) and kelvin probe force microscopy (KPFM) measurements to understand the differences in roughness and work function distribution of in-situ and spin-

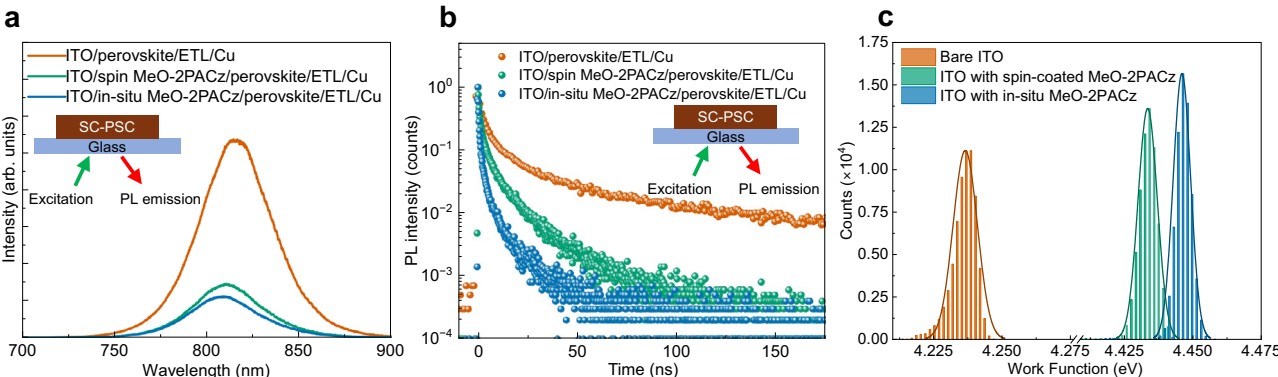

**Fig. 4 | Characterization of perovskite crystals and substrates. a** PL spectra and **b** TRPL decay from the ITO side of SC-PSCs utilizing spin-coated and in-situ MeO-2PACz, and on ITO. PL intensity counts for the TRPL decay plots have been normalized. **c** Work-function distribution on different substrates determined by KPFM. Source data for the plots are provided as a Source Data file.

coated MeO-2PACz coated ITO (Fig. 4c, Supplementary Fig. 10a-10f). The root mean square roughness values for bare ITO, spin-coated MeO-2PACz and in-situ deposited MeO-2PACz were 11.10 nm, 5.74 nm and 6.27 nm respectively. MeO-2PACz coating by spin and in-situ deposition methods results in lowering of surface roughness. The mean work function (WF) of in-situ coated MeO-2PACz was 4.445 V with a standard deviation of 3.2 mV, whereas the mean WF of spin-coated MeO-2PACz was 4.371 V with a standard deviation of 3.5 mV. For reference, the bare ITO had a mean WF of 4.236 V. The higher WF and lower standard deviation observed in the in-situ coated MeO-2PACz indicate improved and more homogeneous molecular coverage compared to the spin-coated film, in line with the CV measurements discussed above.

We also investigated the charge carrier transport characteristics of SC-PSCs by performing transient photocurrent and photovoltage studies (Supplementary Fig. 11a, b). The photocurrent decay time decreased from 0.92 μs in spin-coated MeO-2PACz SC-PSC to 0.24 μs in in-situ deposited MeO-2PACz SC-PSC, suggesting more efficient charge extraction in the latter. Conversely, the photovoltage decay time increased from 2.29 μs to 4.22 μs, suggesting reduced charge carrier recombination in SC-PSCs with in-situ deposited MeO-2PACz compared to their spin-coated counterparts.

All studies discussed thus far suggest that the boost in the performance of the in-situ MeO-2PACz SC-PSC is due to improvement in the ITO/MeO-2PACz/perovskite interfaces. However, there is a possibility that the performance enhancement using ASSC may be due to improved perovskite crystal quality. To probe this hypothesis, we detached perovskite crystals from the spin and in-situ coated ITO substrates, rinsed with acetone to remove any residual MeO-2PACz and measured their PL lifetime ($\tau_{average}$). The crystal grown using ASSC method had a $\tau_{average}$ of 90.32 ns whereas the crystal grown using SSSC method had a $\tau_{average}$ of 76.13 ns (Supplementary Fig. 12), indicating that the ASSC method does improve the quality of the crystal. We attribute this to better wettability of in-situ deposited MeO-2PACz than those with spin-coated MeO-2PACz, as discussed above, which can facilitate crystal growth and its quality.

We evaluated the shelf stability in-situ MeO-2PACz SC-PSC and spin-coated MeO-2PACz SC-PSC (Supplementary Fig. 13). Both devices experience slow spontaneous efficiency enhancement[55]. The spin-coated MeO-2PACz SC-PSC retained 94.8% of its peak PCE for 135 days, whereas the in-situ MeO-2PACz SC-SPC retained 96.7% of its peak PCE even after 150 days. This suggests that in-situ MeO-2PACz devices are more stable than spin-coated MeO-2PACz.

We also assessed the stability of SC-PSCs upon aging in ambient conditions (23 °C, 45 ± 10 % relative humidity). As shown in Supplementary Fig. 14a–d, after 140 hours of exposure to ambient conditions,

the in-situ MeO-2PACz SC-PSC retained 87.5% of its initial efficiency, compared to 74% retention for the spin-coated device. The performance degradation in both cases was primarily due to reductions in $J_{sc}$ and FF, attributed to interfacial hydration caused by moisture exposure[17,56]. Nevertheless, the in-situ devices exhibited lower losses, suggesting that the improved coverage of MeO-2PACz achieved via in-situ deposition offers better protection against moisture-induced interfacial degradation[57].

Finally, we evaluated the operational stability of our SC-PSCs by maximum power point tracking (MPPT) measurements. The in-situ MeO-2PACz SC-PSC retained 92.5% of its initial PCE, while the spin-coated SC-PSC retained 89.6% after 120 hours of operation under MPPT condition (Supplementary Fig. 15). The enhanced operational stability of in-situ MeO-2PACz SC-PSCs over spin-coated devices is attributed to the improved surface coverage achieved through in-situ deposition, which increases wettability and strengthens perovskite adhesion to the substrate[58].

## Discussion

We have shown that an asymmetric ITO/MeO-2PACz substrate stack can be leveraged during SC-ITC to achieve in-situ deposition of MeO-2PACz onto initially uncoated ITO. This process yields a more densely packed and uniformly distributed SAM than conventional spin-coating, thereby enhancing hole extraction, reducing non-radiative recombination, and boosting device performance. As a result, the champion SC-PSCs fabricated via this method attain a PCE of 24.32% which is amongst the highest reported PCE for cesium-free perovskite-based SC-PSCs. These findings underscore the critical impact of substrate-stack design on the interfacial quality of SC-PSCs and highlight in-situ SAM transfer as a powerful tool for overcoming coverage-related limitations. More broadly, this approach provides a promising route toward further improving the efficiency, stability, and scalability of single-crystal perovskite photovoltaics.

## Methods
### Materials
All reagents were purchased from commercial suppliers. [2-(3,6-Dimethoxy-9H-carbazol-9-yl)ethyl]phosphonic acid (MeO-2PACz) was obtained from TCI America. Lead iodide (PbI₂, 99.999%) beads were purchased from Thermo Fisher Scientific. Formamidinium iodide (FAI) and methylammonium iodide (MAI) were acquired from Greatcell Solar Materials. γ-Butyrolactone (GBL), tetrabutylammonium phosphorus hexafluoride (Bu₄NPF₆) and orthodichlorobenzene (oDCB) were sourced from Millipore-Sigma. C₆₀ was supplied by Nano-C, and bathocuproine (BCP) by Lumtec. Copper was obtained from Angstrom Engineering Inc.

## Substrate preparation and MeO-2PACz spin-coating

Indium tin oxide (ITO) coated glass substrates (2 inch × 2 inch) were sequentially cleaned via sonication in detergent, deionized water, acetone, and isopropyl alcohol for 10 minutes each. The substrates were UV-Ozone treated for 15 minutes before MeO-2PACz coating. A 400 µl solution of MeO-2PACz (1 mg/ml in ethanol) was spin-coated at 3000 rpm for 30 seconds followed by annealing at 100 °C for 10 minutes.

## Perovskite solution preparation

A 1.9 M solution of $FA_{0.6}MA_{0.4}PbI_3$ in GBL was prepared by dissolving 1.14 mmoles of FAI, 0.76 mmoles of MAI and 1.9 mmoles of $PbI_2$ in 1 ml of GBL. The mixture was heated to 50 °C and continuously stirred until complete dissolution, resulting in a clear yellow solution.

## Thin perovskite crystal growth procedure

MeO-2PACz-coated substrates were heated to 50 °C. Then, 60 µL of the perovskite solution was dropped onto the substrate. In the case of a symmetric substrate stack configuration (SSSC), another MeO-2PACz-coated substrate was placed on top to form a sandwich structure. For the asymmetric substrate stack configuration (ASSC), an uncoated ITO substrate was used instead. The temperature was gradually increased to 75 °C at a rate of 15 °C/hr, followed by a further increase to 130 °C at a slower rate of 3 °C/hr. Crystal growth was allowed to proceed at 130 °C for 3 hours. Afterward, the substrates were cooled and carefully separated using a razor. Any remaining solution was swiftly removed using Kimwipes.

## Device fabrication

$C_{60}$ (20 nm) and BCP (6 nm) were thermally evaporated onto the perovskite crystal surface at a rate of 0.1 Å/s. The edges of the single crystals were then masked with Kapton tape. Finally, Cu (80 nm) was deposited at 1 Å/s to complete the solar cells. To ensure accurate measurement of the active area, the glass side of each crystal was outlined with black tape. The area was measured using a microscope.

## Material and device characterization

The X-ray Diffraction (XRD) analyses were conducted using a PANalytical Empyrean system equipped with a Cu source ($K_\alpha$, λ = 1.5406 Å). Steady state photoluminescence (PL) spectroscopy was carried out on a RENISHAW inVia confocal Raman microscope. Time-resolved photoluminescence (TRPL) measurements were taken using an Edinburgh Instruments OB920 Single Photon Counting system, with excitation provided by a 510 nm pulsed laser diode. Current density-voltage (J-V) characteristics of the devices were recorded using a Keithley 2400 source meter and a G2V Pico Class AAA LED and Abet Technologies, Sun 3000 under Air Mass (AM) 1.5 G illumination (100 mW cm⁻²), with light intensity calibrated using a silicon reference solar cell. External Quantum Efficiency (EQE) was measured using QEX10 Spectral Response Measurement System, PV Measurements, Inc. X-ray photoelectron spectroscopy (XPS) experiments were performed using Thermofisher Scientific K-Alpha XPS system. Fourier Transformed Infrared (FTIR) Spectroscopy was performed using Cary 630 spectrometer using Ge attenuated total reflectance module. Atomic Force Microscopy (AFM) and Kelvin Probe Force Microscopy (KPFM) images were acquired using the MFP-3D SPM system from Asylum Research. The tip used for KPFM measurements was calibrated using highly oriented pyrolytic graphite sample. The surface potential map generated by KPFM measurements was used to calculate work function distribution. Transient photovoltage and transient photocurrent measurements were performed using Opotek INC. RADIANT SE 355 LD, a nanosecond pulsed laser operating at 500 nm and Tektronix digital oscilloscope. During these measurements, the devices were illuminated with light beam having 1-sun intensity. Contact angles measurements were done using Holmarc's Contact Angle Meter

(HO-IAD-CAM-01). Operational stability of encapsulated devices was evaluated by conducting maximum power point tracking (MPPT) measurements using a custom-designed LED simulator inside a nitrogen-filled glovebox atmosphere with temperature around 55 °C. The operational stability data was recorded using an Ossila source meter.

## Cyclic voltammetry measurement and MeO-2PACz surface density calculation

The cyclic voltammetry (CV) measurements were performed using Gamry 1010E potentiostat. The measurements were carried out at ambient temperature under an inert atmosphere, employing a three-electrode setup with a bare or MeO-2PACz-coated ITO as the working electrode, a platinum wire as the counter electrode, and an Ag wire pseudo-reference electrode, separated from the solution via a porous frit. The supporting electrolyte was a 0.1 M $Bu_4NPF_6$ solution in oDCB. Ferrocene/ferrocenium redox potentials were measured for calibration. Each sample's CV was recorded at various scan rates.

The number of molecules adsorbed on a conducting surface can be calculated from the CV measurements using the following equation:

$$i_p = \frac{n^2 F^2}{4RTN_A} A\Gamma^* \nu \tag{1}$$

where $i_p$ is the oxidative peak current in amperes, F is the Faraday's constant (96,485.33 Cmol⁻¹), $n$ is the number of electrons transferred, R is the universal gas constant (8.314 JK⁻¹mol⁻¹), $T$ is the temperature in kelvin, $N_A$ (6.023 × 10²³ mol⁻¹) is the Avogadro constant, $A$ is the electrode surface area in square centimeters, $\Gamma^*$ is the molecular surface density per square centimeter and $\nu$ is the scan rate in volts per second. By utilizing the value of the slope of the plot between $i_p$ and $\nu$, $\Gamma^*$ can be calculated.

## Computational details

The optimizations and frequency computations were performed using the Vienna Ab initio Simulation Package (VASP)[59,60]. The projector augmented wave (PAW) method was used to describe the ion-electron interactions[61,62]. The generalized-gradient approximation (GGA) with the Perdew-Burke-Ernzerhof (PBE) functional[63,64] was used as the exchange-correlation functional. DFT-D3 Grimme's scheme method[65] for van der Waals correction was adopted for the slab calculations. The kinetic energy cutoff for plane wave expansions was set to 450 eV. Four-layer ITO [111] slabs were constructed to simulate the surface structure with two bottom layers fixed, and a vacuum of at least 15 Å was used to avoid the interaction of the adjacent images. The reciprocal space was sampled using the Monkhorst-Pack 3×3×1 k-point mesh. All the geometric structures were optimized until the maximal components of forces converged to within 3×10⁻³ eV·Å⁻¹. The binding energies of MeO-2PACz molecule on ITO [111] surface were calculated at 373.15, 383.15, 393.15 and 403.15 K respectively. Corrections of zero-point energy (ZPE) and enthalpy energy H(T) at different temperatures were calculated by VASPKIT software[66].

The binding energies ($E_b$) were calculated by the following equation:

$$E_b = E_{Total} - E_{ITO} - E_{MeO-2PACz} \tag{2}$$

Here, the $E_{Total}$, $E_{ITO}$, and $E_{MeO-2PACz}$ are the energies of ITO [111] slab binding with MeO-2PACz molecule, the clean ITO [111] slab, and the isolated MeO-2PACz molecule, respectively. The DFT simulated structure is shown in Fig. 3d, and the corresponding atomic coordinates are provided in the Supplementary Information (Supplementary Data 1). The Crystallographic Information File data for this DFT simulated structure is also provided in an Excel worksheet included in the Supplementary Information (Supplementary Data 2).

**Disclosure of artificial intelligence (AI) assisted technologies used during manuscript writing**

ChatGPT was occasionally used to improve grammar and sentence structure during the preparation of this manuscript. The authors reviewed all content and take full responsibility for the final manuscript.

**Reporting summary**

Further information on research design is available in the Nature Portfolio Reporting Summary linked to this article.

## Data availability

Source data are provided with this paper.

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

## Acknowledgements

We thank the Natural Sciences and Engineering Research Council of Canada, and Mitacs (ALLRP 561355-20; IT26591) for their financial contributions. This publication in part is based upon work supported by King Abdullah University of Science and Technology (ORFS-2022-CRG11-5345.3) and NSERC Alliance International Grant (ALLRP 586212 - 23). M.I.S. thanks the Canada Research Chairs Program (CRC-2024-00264), the Canada Foundation for Innovation (40326), and the B.C. Knowledge Development Fund (806169) for their contributions to infrastructure support. We acknowledge and respect the Lək̓ʷəŋən (Songhees and Xʷsepsəm/Esquimalt) Peoples on whose territory the University of Victoria stands, and the Lək̓ʷəŋən and W̱SÁNEĆ Peoples whose historical relationships with the land continue to this day.

## Author contributions

V.Y. and K.A. contributed equally to this work. V.Y., K.A., O.M.B., and M.I.S. conceived the idea and initiated the project. V.Y. and K.A. were responsible for the fabrication of the solar cells and performance evaluation. Y.X. and O.F.M. led the DFT study. A.A. and S.Q. conducted the AFM and KPFM measurements. C.T. performed the CV measurements under the supervision of H.L.B. S.D. carried out the TPV and TPC measurements. V.Y. and M.I.S. drafted the initial version of the manuscript. M.N.L., D.Z., and P.M. contributed to manuscript preparation. All authors reviewed and contributed to the final manuscript. O.M.B. and M.I.S. co-supervised the project.

## Competing interests

The authors declare no competing interests.
