## [Transparent Peer Review file · Nature Communications]

In-Situ Self-Assembly of Hole Transport Monolayer During Crystallization for Efficient Single-Crystal Perovskite Solar Cells

Corresponding Author: Professor Makhsud Saidaminov

Version 0:

Reviewer comments:

Reviewer #1

(Remarks to the Author)

In this manuscript, the authors employed self-assembled monolayer (SAM) in an asymmetric substrate stack configuration to fabricate single-crystal perovskite solar cells (SC-PSCs). During the space-confined inverse temperature crystallization growth process, the SAM molecules migrated from SAM-coated substrate to vacant substrate, achieving dense coverage of SAM on vacant substrate. As a result, an enhanced power conversion efficiency of 24.32% was achieved for SC-PSCs. However, the specific mechanism is still vague. In addition, much higher PCE of SC-PSCs as 25.8% was achieved by strengthening crystal/substrate contact (Angew. Chem. Int. Ed. 2025, e202500947), and the PCE of 24.32% in this manuscript shows less competitiveness. In conclusion, I think this manuscript needs further improvement and cannot recommend the publication on Nature Communications at its present form.

1. Why were the different sizes of perovskite crystals obtained in different substrate in Fig 1c? And more size distribution data is needed here to illustrate the regulation of crystallization for asymmetric substrate stack configuration.
2. The authors should explain why the SAM coated on the substrate migrated to the untreated substrate and formed denser coverage on vacant substrate, rather than redistributed on the original substrate. And why lower coverage of SAM in substrates from SSSC were obtained?
3. Please include the different effects of self-assembled molecules on asymmetric substrate stack configuration. More systematic study needs to be conducted.

Reviewer #2

(Remarks to the Author)

- (1) This manuscript proposes an in-situ self-assembly SAM strategy using asymmetric substrate stack structure (ASSC) in the spatially constrained inversion crystallization (SC-ITC) process to improve the interface of SC-PSCs. The results indicate that it is effective in improving device performance. However, this article mainly studied the changes in the SAM layer, but it cannot be ruled out that the ASSC method may also improve the quality of the perovskite single crystal itself. Therefore, it is necessary to further characterize the quality of the single crystal to verify whether the performance improvement is solely attributed to the modification of the SAM layer.
- (2) Although AFM measurement is provided, the root mean square (RMS) roughness of the sample is not reported. Suggest calculating and comparing RMS values to quantitatively evaluate surface roughness differences between samples.
- (3) All samples were stored in nitrogen filled glove boxes, but their stability under different environmental conditions was not checked. Suggest conducting humidity/oxygen aging tests to improve the relevance of this study for practical applications.

Reviewer #3

(Remarks to the Author)

In this manuscript, the authors have demonstrated a novel SAM deposition strategy by using an asymmetric substrate stack

configuration during crystal growth, achieving a highly efficiency of 24.32% for FA0.6MA0.4PbI3 single-crystal solar cells. Compared to spin-coated SAM layer, the in-situ deposition strategy achieves a denser and more homogeneous SAM distribution, resulting in superior crystal attachment, enhanced hole extraction and reduced charge recombination. Besides, the optimized interface contact also improves the storage stability of single-crystal devices. Overall, this work is important and innovative for the field of single-crystal PSCs and will be of wide interest to the readers. There are, however, a few minor comments that I believe should be addressed before this paper is published.

1. In Figure 1c, some impurities were observed on all the obtained crystal surfaces. The authors should present more detailed explanation, and provide higher-resolution crystal images for clearer observation.
2. For the statistical distribution of photovoltaic parameter, the solar cells based on ITO from ASSC showed a wider PCE distribution than that of control devices. Can the authors clarify this aspect?
3. The authors evaluated the carrier extraction at the crystal interface by PL and TRPL characterization. However, carrier transport properties in single-crystal devices should also be conducted, such as transient photocurrent measurement.
4. In addition to the storage stability of FA0.6MA0.4PbI3 single-crystal devices, the operational stability of devices based on spin-coated MeO-2PACz and in-situ MeO-2PACz should be supplied as a comparison.
5. Recently, Liu et al. obtained a new efficiency benchmark for single-crystal PSCs (Angew. Chem. Int. Ed. 2025, e202500947; doi.org/10.1002/anie.202500947), which should be cited and described in the introduction section.

Version 1:

Reviewer comments:

Reviewer #1

(Remarks to the Author)

The authors have revised the manuscript carefully according to the reviewers' comments and revision opinions. Now the revised manuscript can be accepted for publication at its present form.

Reviewer #2

(Remarks to the Author)

All suggested revisions have been satisfactorily implemented. The current version is now acceptable for publication.

Reviewer #3

(Remarks to the Author)

The authors have adequately revised their manuscript in response to my comments. I recommend the publication of this work.

Point-by-point actions in response to reviewers' comments on Manuscript ID: NCOMMS-25-18903

We sincerely appreciate the reviewers' insightful suggestions, which have greatly contributed to improving the quality of our work. We provide below a detailed account of the revisions made in response.

Reviewer #1 (Remarks to the Author):

In this manuscript, the authors employed self-assembled monolayer (SAM) in an asymmetric substrate stack configuration to fabricate single-crystal perovskite solar cells (SC-PSCs). During the space confined inverse temperature crystallization growth process, the SAM molecules migrated from SAM coated substrate to vacant substrate, achieving dense coverage of SAM on vacant substrate. As a result, an enhanced power conversion efficiency of 24.32% was achieved for SC-PSCs. However, the specific mechanism is still vague. In addition, much higher PCE of SC-PSCs as 25.8% was achieved by strengthening crystal/substrate contact (Angew. Chem. Int. Ed. 2025, e202500947), and the PCE of 24.32% in this manuscript shows less competitiveness. In conclusion, I think this manuscript needs further improvement and cannot recommend the publication on Nature Communications at its present form.

Comment 1:

Why were the different sizes of perovskite crystals obtained in different substrate in Fig 1c? And more size distribution data is needed here to illustrate the regulation of crystallization for asymmetric substrate stack configuration.

Response:

As requested by the reviewer, we have added a new figure (Figure S1) which provides detailed perovskite crystal size distribution data for different substrates. This figure shows that the crystals grown on ITO from ASSC, on average, are larger in size than crystals on MeO-2PACz from SSSC and ASSC. We now write on page 5 of the revised manuscript:

Figure S1 presents the statistical distribution of perovskite crystal sizes across different substrate types. Crystals grown on the ITO exhibited a larger average size compared to those on ITO/MeO-2PACz in both SSSC and ASSC configurations.

Figure S1. Perovskite crystal size distribution.

Box-and-whisker plots showing size distribution of perovskite crystals for each substrate type. The box boundaries represent the first and third quartiles (25th and 75th percentiles), the horizontal line within each box indicates the median, and the hollow square marks the average crystal size. The whiskers extending above and below each box show the range of values that fall within the normal spread.

To understand why in-situ deposited MeO-2PACz promotes crystal growth, we performed contact angle measurement, which showed that the substrates with in-situ deposited MeO-2PACz exhibit better wettability of perovskite solution than spin-coated MeO-2PACz does (newly added Figure S9). Better wettability of perovskite solution was shown to promote crystal growth (*ACS Energy Lett.* 2023, 8, 2, 950). The increased wettability of transferred MeO-2PACz substrates hence explain why crystal grew larger on these substrates compared to spin-coated MeO-2PACz. We now write on page 11 of the revised manuscript:

We investigated the differences between spin-coated MeO-2PACz and in-situ deposited MeO-2PACz by contact angle measurements using a supersaturated FA_{0.6}MA_{0.4}PbI₃ perovskite solution in gamma-butyrolactone (Figure S9). The bare ITO substrate had a contact angle of 62.5° suggesting poor wettability of perovskite solution. On the other hand, the contact angle for spin-coated and in-situ deposited MeO-2PACz substrates were sharp, 13.4° and 11.3°, respectively. The lower contact for in-situ MeO-2PACz indicates better perovskite solution wettability compared to spin-coated MeO-2PACz. The improved wettability is due to enhanced surface coverage obtained using the in-situ deposition technique. The increased wettability also explains why crystals on ITO from ASSC were, on average, larger compared to crystals on MeO-2PACz from SSSC since better wettability can promote crystal growth³⁴.

Figure S9. Contact angle measurements.

Contact angle measurements of bare ITO, spin-coated MeO-2PACz and in-situ coated MeO-2PACz. The measurements were performed using supersaturated $\text{FA}_{0.6}\text{MA}_{0.4}\text{PbI}_3$ perovskite solution in gamma-butyrolactone (GBL).

Comment 2:

The authors should explain why the SAM coated on the substrate migrated to the untreated substrate and formed denser coverage on vacant substrate, rather than redistributed on the original substrate. And why lower coverage of SAM in substrates from SSSC were obtained?

Response:

Using density functional theory calculations, we now provide an in-depth mechanism for in-situ MeO-2PACz deposition. We now write on page 10 of the revised manuscript:

To gain insight into the in-situ deposition process, we performed density functional theory (DFT) calculations to examine the interaction between MeO-2PACz and ITO. The XRD pattern of our ITO showed highest diffraction intensity off (222) crystallographic plane, which is a higher-order diffraction of (111) plane (Figure S7). Therefore, we modeled the InO-terminated ITO (111) surface. The results indicate that MeO-2PACz attaches to ITO (111) via two types of bonds: a P=O–In bond and a P–O–In bond (Figure 3d). The latter P–O–In bond forms when a phosphonic acid O–H group in MeO-2PACz dissociates, yielding a deprotonated oxygen (O^-) that binds to an indium atom and a proton (H^+) that bonds to an oxygen on the ITO surface. We further calculated binding energies at 373.15 K (100 °C), 383.15 K (110 °C), 393.15 K (120 °C), and 403.15 K (130 °C) (spanning the typical substrate

temperatures during perovskite crystallization). Figure 3e shows that the binding energy of MeO-2PACz with ITO decreases as the temperature increases. During the SC-ITC process, the bottom substrate (in contact with the hot plate) is at a higher temperature than the top substrate (exposed to the cooler atmosphere, Figure S8). As a result, MeO-2PACz is expected to bind more strongly to the cooler top ITO surface than to the hotter bottom ITO.

Figure 3f illustrates a possible mechanism for the transfer of MeO-2PACz during the crystallization process. The concentration of MeO-2PACz in the ethanol used for spin-coating exceeds the critical micelle formation concentration, causing the molecules to form micelles in the solution; these aggregated structures remain intact after spin-coating, leading to non-uniform coverage and reduced ITO surface coverage⁴². During SC-ITC using an ITO/MeO-2PACz ASSC, the loosely bound MeO-2PACz molecules dissolve into the perovskite solution⁴⁴. However, since the binding energy of MeO-2PACz with cooler top ITO substrate is higher than that of the hotter bottom ITO substrate, the dissolved MeO-2PACz molecules preferentially adsorb onto the top uncoated ITO substrate resulting in in-situ deposition. As this process is slower than spin-coating, in-situ deposition process during crystallization likely allows the MeO-2PACz molecules to organize into a more orderly monolayer, resulting in improved surface coverage.

Comparing the MeO-2PACz coverage for different substrates in SSSC and ASSC provides further insights into the in-situ deposition process. In SSSC, a greater total amount of MeO-2PACz is available to dissolve into the perovskite solution, since both top and bottom MeO-2PACz coated substrates contribute loosely bound SAM molecules. This higher dissolved concentration leads to more re-adsorption and slightly enhanced overall coverage on the spin-coated substrate in the SSSC, compared to the spin-coated substrate in ASSC. Interestingly, MeO-2PACz coverage on the top ITO substrate from ASSC exceeds that of the spin-coated substrates used in SSSC. This is probably because the spin-coating results in defects which cannot be healed by re-adsorption of MeO-2PACz dissolved in perovskite solution thereby limiting its surface coverage. In contrast, the initially uncoated ITO surface in ASSC enables fresh, more ordered adsorption of MeO-2PACz during crystallization, resulting in a more uniform and defect-free monolayer.

Figure 3. **d** Side-view DFT-simulated schematic illustrating the anchoring of MeO-2PACz on InO-terminated ITO (111) surface. **e** ITO/MeO-2PACz binding energies at different temperatures.

Figure S7. **XRD of ITO.**

XRD pattern of ITO coated glass substrate.

Figure S8. **Temperature difference between top and bottom substrate.**

Measure temperature of top and bottom substrates during crystallization.

Comment 3:

Please include the different effects of self-assembled molecules on asymmetric substrate stack configuration. More systematic study needs to be conducted.

Response:

In response to all the reviewers' comments, we have conducted additional investigations to systematically study the effects of self-assembled molecules in asymmetric substrate stack configuration. Specifically, we performed the following additional experiments:

1. Crystal size analysis, as discussed above in response to your comment 1.
2. DFT calculations to understanding in-situ deposition mechanism, as discussed above in response to your comment 2.
3. Contact angle measurements on different substrates, as discussed in response to your comment 1
4. Transient photocurrent and photovoltage measurement of single crystal perovskite solar cell (Page 12):

We also investigated the charge carrier transport characteristics of SC-PSCs by performing transient photocurrent and photovoltage studies (Figure S11). The photocurrent decay time decreased from 0.92 μs in spin-coated MeO-2PACz SC-PSC to 0.24 μs in in-situ deposited MeO-2PACz SC-PSC, suggesting more efficient charge extraction in the latter. Conversely, the photovoltage decay time increased from 2.29 μs to 4.22 μs , suggesting reduced charge carrier recombination in SC-PSCs with in-situ deposited MeO-2PACz compared to their spin-coated counterparts.

Figure S11. TPC and TPV measurements of SC-PSCs.

a TPC and **b** TPV of SC-PSCs with spin and in-situ deposited MeO-2PACz as hole transport layer.

5. Time-resolved photoluminescence lifetimes measurements to evaluate the effect of asymmetric configuration on the crystal quality (Page 12):

All studies discussed thus far suggest that the boost in the performance of the in-situ MeO-2PACz SC-PSC is due to improvement in the ITO/MeO-2PACz/perovskite interfaces. However, there is a possibility that the performance enhancement using ASSC may be due to improved perovskite crystal quality. To probe this hypothesis, we detached perovskite crystals from the spin and in-situ coated ITO substrates, rinsed with acetone to remove any residual MeO-2PACz and measured their PL lifetime (τ_{average}). The crystal grown using ASSC method had a τ_{average} of 90.32 ns whereas the crystal grown using SSSC method had a τ_{average} of 76.13 ns (Figure S12), indicating that the ASSC method does improve the quality of the crystal. We attribute this to better wettability of in-situ deposited MeO-2PACz than those with spin-coated MeO-2PACz, as discussed above, which can facilitate crystal growth and its quality.

Figure S12. TRPL of the detached crystals.

TRPL of perovskite crystals detached from substrates having spin and in-situ coated MeO-2PACz.

6. Ambient stability of symmetric and asymmetric single crystal perovskite solar cells (Page 13):

We also assessed the stability of SC-PSCs upon aging in ambient conditions (23 °C, 45 ± 10 % relative humidity). As shown in Figure S14, after 140 hours of exposure to ambient conditions, the in-situ MeO-2PACz SC-PSC retained 87.5% of its initial efficiency, compared to 74% retention for the spin-coated device. The performance degradation in both cases was primarily due to reductions in J_{sc} and FF, attributed to interfacial hydration caused by moisture exposure^{58,59}. Nevertheless, the in-situ devices exhibited lower losses, suggesting that the improved coverage of MeO-2PACz achieved via in-situ deposition offers better protection against moisture-induced interfacial degradation⁶⁰.

Figure S14. Stability of SC-PSCs under ambient conditions.

Evolution of (a) V_{oc} , (b) J_{sc} (c) fill factor and (d) PCE of spin-coated and in-situ deposited SC-PSCs upon storage in ambient conditions (23 °C, 45 ± 10 % relative humidity).

7. Operational stability of symmetric and asymmetric single crystal perovskite solar cells (Page 13):

Finally, we evaluated the operational stability of our SC-PSCs by maximum power point tracking (MPPT) measurements. The in-situ MeO-2PACz SC-PSC retained 92.5% of its initial PCE, while the spin-coated SC-PSC retained 89.6% after 120 hours of operation under MPPT condition (Figure S15). The enhanced operational stability of in-situ MeO-2PACz SC-PSCs over spin-coated devices is attributed to the improved surface coverage achieved through in-situ deposition, which increases wettability and strengthens perovskite adhesion to the substrate⁶¹.

Figure S15. Operational stability of SC-PSCs.

Operational stability of spin-coated and in-situ deposited MeO-2PACz SC-PSCs determined by MPP tracking. Measurements were performed using unencapsulated SC-PSCs inside N₂ filled glovebox at around 55 °C.

Reviewer #2 (Remarks to the Author):

Comment 1:

This manuscript proposes an in-situ self-assembly SAM strategy using asymmetric substrate stack structure (ASSC) in the spatially constrained inversion crystallization (SC-ITC) process to improve the interface of SC-PSCs. The results indicate that it is effective in improving device performance. However, this article mainly studied the changes in the SAM layer, but it cannot be ruled out that the ASSC method may also improve the quality of the perovskite single crystal itself. Therefore, it is necessary to further characterize the quality of the single crystal to verify whether the performance improvement is solely attributed to the modification of the SAM layer.

Response:

The reviewer asks whether our approach might also impact the quality of perovskite crystal. We probed this hypothesis, and now write on page 12 of the revised manuscript:

All studies discussed thus far suggest that the boost in the performance of the in-situ MeO-2PACz SC-PSC is due to improvement in the ITO/MeO-2PACz/perovskite interfaces. However, there is a possibility that the performance enhancement using ASSC may be due to improved perovskite crystal quality. To probe this hypothesis, we detached perovskite crystals from the spin and in-situ coated ITO substrates, rinsed with acetone to remove any residual MeO-2PACz and measured their PL lifetime (τ_{average}). The crystal grown using ASSC method had a τ_{average} of 90.32 ns whereas the crystal grown using SSSC method had a τ_{average} of 76.13 ns (Figure S12), indicating that the ASSC method does improve the quality of the crystal. We attribute this to better wettability of in-situ deposited MeO-2PACz than those with spin-coated MeO-2PACz, as discussed above, which can facilitate crystal growth and its quality.

Figure S12. TRPL of the detached crystals.

TRPL of perovskite crystals detached from substrates having spin and in-situ coated MeO-2PACz.

Comment 2:

Although AFM measurement is provided, the root mean square (RMS) roughness of the sample is not reported. Suggest calculating and comparing RMS values to quantitatively evaluate surface roughness differences between samples.

Response:

As suggested, we now write on page 12 of the revised manuscript:

The root mean square roughness values for bare ITO, spin-coated MeO-2PACz and in-situ deposited MeO-2PACz were 11.10 nm, 5.74 nm and 6.27 nm respectively. MeO-2PACz coating by spin and in-situ deposition methods results in lowering of surface roughness.

Comment 3:

All samples were stored in nitrogen filled glove boxes, but their stability under different environmental conditions was not checked. Suggest conducting humidity/oxygen aging tests to improve the relevance of this study for practical applications.

Response:

We assessed the stability of devices upon aging in ambient conditions (23 °C, 45 ± 10% relative humidity). The results and discussion of these measurements have been added on page 13 of the revised manuscript:

We also assessed the stability of SC-PSCs upon aging in ambient conditions (23 °C, 45 ± 10 % relative humidity). As shown in Figure S14, after 140 hours of exposure to ambient conditions, the in-situ MeO-2PACz SC-PSC retained 87.5% of its initial efficiency, compared to 74% retention for the spin-coated device. The performance degradation in both cases was primarily due to reductions in J_{sc} and FF, attributed to interfacial hydration caused by moisture exposure^{58,59}. Nevertheless, the in-situ devices exhibited lower losses, suggesting that the improved coverage of MeO-2PACz achieved via in-situ deposition offers better protection against moisture-induced interfacial degradation⁶⁰.

Figure S14. Stability of SC-PSCs under ambient conditions.

Evolution of (a) V_{oc} , (b) J_{sc} (c) fill factor and (d) PCE of spin-coated and in-situ deposited SC-PSCs upon storage in ambient conditions (23 °C, 45 ± 10 % relative humidity).

Reviewer #3 (Remarks to the Author):

In this manuscript, the authors have demonstrated a novel SAM deposition strategy by using an asymmetric substrate stack configuration during crystal growth, achieving a highly efficiency of 24.32% for FA0.6MA0.4PbI3 single-crystal solar cells. Compared to spin-coated SAM layer, the in-situ deposition strategy achieves a denser and more homogeneous SAM distribution, resulting in superior crystal attachment, enhanced hole extraction and reduced charge recombination. Besides, the optimized interface contact also improves the storage stability of single-crystal devices. Overall, this work is important and innovative for the field of single-crystal PSCs and will be of wide interest to the readers. There are, however, a few minor comments that I believe should be addressed before this paper is published.

Comment 1:

In Figure 1c, some impurities were observed on all the obtained crystal surfaces. The authors should present more detailed explanation and provide higher-resolution crystal images for clearer observation.

Response:

We have now included a detailed explanation of these surface features on page 5 of the manuscript:

The perovskite crystals shown in Figure 1c appear to have spots on surfaces which we attribute to be defects resulting from residual perovskite growth solution left on the crystal after crystallization was terminated (Figure S2). As this residual solution cooled rapidly, it lost supersaturation, leading to localized dissolution of the perovskite. The use of surfactants such as cetyltrimethylammonium chloride has been shown to prevent such surface damage⁴⁹. However, we chose not to use this surfactant in this work to solely focus on the effects of substrate stack configurations.

Figure S2. Optical micrograph of crystal surface.

Optical micrograph of $\text{FA}_{0.6}\text{MA}_{0.4}\text{PbI}_3$ single crystal surface zooming on spots observed in the pictures of perovskite crystals.

Comment 2:

For the statistical distribution of photovoltaic parameter, the solar cells based on ITO from ASSC showed a wider PCE distribution than that of control devices. Can the authors clarify this aspect?

Response:

In the ITO/MeO-2PACz ASSC configuration, MeO-2PACz is deposited in-situ on the bare ITO substrate. Because this transfer occurs spontaneously during crystallization, it can lead to higher degree of variation in the quality of in-situ deposited MeO-2PACz films resulting in slightly wider PCE distribution compared to MeO-2PACz SSSC devices. However, the average PCE of the ITO from ASSC devices is higher than that of control MeO-2PACz SSSC devices. This suggests that despite higher degree of variability, the quality of in-situ deposited MeO-2PACz film is superior to that of spin-coated MeO-2PACz film.

Comment 3:

The authors evaluated the carrier extraction at the crystal interface by PL and TRPL characterization. However, carrier transport properties in single-crystal devices should also be conducted, such as transient photocurrent measurement.

Response:

As suggested, we investigated carrier transport properties in our single-crystal devices using transient photocurrent (TPC) and transient photovoltage (TPV) measurements, and now write on page 12:

We also investigated the charge carrier transport characteristics of SC-PSCs by performing transient photocurrent and photovoltage studies (Figure S11). The photocurrent decay time decreased from 0.92 μs in spin-coated MeO-2PACz SC-PSC to 0.24 μs in in-situ deposited MeO-2PACz SC-PSC, suggesting more efficient charge extraction in the latter. Conversely, the photovoltage decay time increased from 2.29 μs to 4.22 μs , suggesting reduced charge carrier recombination in SC-PSCs with in-situ deposited MeO-2PACz compared to their spin-coated counterparts.

Figure S11. TPC and TPV measurements of SC-PSCs.

a TPC and b TPV of SC-PSCs with spin and in-situ deposited MeO-2PACz as hole transport layer.

Comment 4:

In addition to the storage stability of FA0.6MA0.4PbI3 single-crystal devices, the operational stability of devices based on spin-coated MeO-2PACz and in-situ MeO-2PACz should be supplied as a comparison.

Response:

We have measured the operational stability of SC-PSCs by doing maximum power point tracking (MPPT) measurement. The results of these measurements have been added to the page 13 of the manuscript:

Finally, we evaluated the operational stability of our SC-PSCs by maximum power point tracking (MPPT) measurements. The in-situ MeO-2PACz SC-PSC retained 92.5% of its initial PCE, while the spin-coated SC-PSC retained 89.6% after 120 hours of operation under MPPT condition (Figure S15). The enhanced operational stability of in-situ MeO-2PACz SC-PSCs over spin-coated devices is attributed to the improved surface coverage achieved through in-situ deposition, which increases wettability and strengthens perovskite adhesion to the substrate⁶¹.

Figure S15. Operational stability of SC-PSCs.

Operational stability of spin-coated and in-situ deposited MeO-2PACz SC-PSCs determined by MPP tracking. Measurements were performed using unencapsulated SC-PSCs inside N₂ filled glovebox at around 55 °C.

Comment 5:

Recently, Liu et al. obtained a new efficiency benchmark for single-crystal PSCs (Angew. Chem. Int. Ed. 2025, e202500947; doi.org/10.1002/anie.202500947), which should be cited and described in the introduction section.

Response:

As suggested, we have updated the introduction with the new efficiency benchmark for single-crystal PSCs (Angew. Chem. Int. Ed. 2025, e202500947), which is now cited under ref. 32 in the revised manuscript.